# Barriers and practices in pain management for intubated patients: A study of critical care nurses in Southern West Bank hospitals

**Ibrahim Salim[1], Moath Abu Ejheisheh[2], Ahmad Ayed [3], Ibrahim Aqtam [4]\*,
Ahmad Batran[2]**

1 Al Mezan Specialized Hospital, Hebron, Palestine, 2 Faculty of Allied Medical Sciences, Department of
Nursing, Palestine Ahliya University, Bethlehem, Palestine, 3 Faculty of Nursing, Arab American University,
Palestine, 4 Department of Nursing, Ibn Sina College for Health Professions, Nablus University for
Vocational and Technical Education, Nablus, Bethlehem, Palestine

\* ibrahim.aqtam@nu-vte.edu.ps

## Abstract

### Introduction

Effective pain management is vital for intubated patients in intensive care units, as these
individuals cannot verbally communicate their discomfort. The knowledge, attitudes, practices, and perceived obstacles of nurses are critical factors that influence successful pain
management.

### Objective

This study aimed to evaluate the knowledge, attitudes, practices, and perceived obstacles
faced by critical care nurses regarding pain management in intubated patients within hospitals located in Southern West Bank.

### Methods

A cross-sectional survey was carried out with 199 critical care nurses utilizing the Nurses'
Knowledge and Attitudes Survey Regarding Pain, alongside a modified tool for assessing
perceived barriers to pain management. The data collected were analyzed to pinpoint deficiencies in knowledge and barriers impacting effective pain management strategies.

### Results

The analysis showed that 192 out of 199 nurses (96.5%) lacked sufficient knowledge
regarding pain management. System-related barriers, including the lack of standardized
protocols and ineffective communication with physicians, were frequently identified as
obstacles, averaging a score of 2.41 out of 3. Additionally, nurse-related barriers comprised insufficient time for providing non-pharmacological interventions (73.9%) and a lack
of confidence in utilizing assessment tools (43.7%). Patient-related issues, such as difficulties in communication (72.4%) and hesitancy to report pain (58.8%), were also noted.

Sciences and Education University, JORDAN

**Peer Review History:** PLOS recognizes the
benefits of transparency in the peer review
process; therefore, we enable the publication
of all of the content of peer review and
author responses alongside final, published
articles. The editorial history of this article is
available here: https://doi.org/10.1371/journal.
pone.0320430

**Data availability statement:** All relevant data are within the manuscript and its Supporting Information files.

**Funding:** The author(s) received no specific funding for this work.

**Competing interests:** The authors have declared that no competing interests exist.

A notable difference in knowledge scores was observed based on prior education in pain management, with those having received training (41.7%) performing better.

## Conclusion

The findings underscore the need for focused educational initiatives, the establishment of standardized protocols for pain management, and enhanced interdisciplinary communication. These improvements are essential for optimizing pain management practices and ensuring better patient outcomes. The results suggest that addressing knowledge gaps and systemic barriers could lead to significant enhancements in patient care and health policies.

## Introduction

Effective pain management is a critical component of patient care, particularly in intensive care units (ICUs), where intubated patients often endure significant pain without the ability to communicate their discomfort, a challenge observed globally across critical care settings [1,2]. Proper pain management improves patient outcomes, prevents complications, and enhances overall quality of life [3,4]. Nurses play a key role in assessing and managing pain in these environments, with their knowledge, attitudes, and practices significantly influencing the quality of care provided [5]. However, evidence suggests that nurses in ICUs often lack sufficient knowledge and training in pain management, leading to inconsistent and inadequate patient care [6,7].

Despite its importance, research indicates that critical care nurses frequently struggle with pain assessment and treatment due to knowledge gaps, limited resources, and inconsistent institutional protocols [8]. In high-resource settings, guidelines for pain management are well established, but in low-resource areas, including the Southern West Bank, nurses often rely on individual clinical judgment rather than standardized protocols [9,10]. Barriers such as inadequate educational resources, the absence of clear guidelines, and time constraints often result in suboptimal pain relief, negatively impacting patient recovery and satisfaction [11,12]. While international guidelines advocate for comprehensive pain management strategies, adherence varies widely, particularly in low-resource settings such as those in the Southern West Bank [13].

Existing research in the Southern West Bank remains limited regarding pain management for intubated patients. Most studies focus on general pain management in ICUs without addressing the specific challenges faced by nurses managing non-verbal patients [14,15]. Furthermore, prior studies fail to comprehensively examine systemic, nurse-related, and patient-related barriers in this region [16]. Addressing pain in this patient demographic presents distinct challenges due to communication difficulties, compounded by the lack of standardized protocols and inconsistencies in nursing education [17]. Additionally, previous studies in this region have not explored the impact of selection bias in research on nurses' pain management practices, further limiting the generalizability of findings [18].

By identifying these gaps, this study aims to provide new insights into the challenges and solutions for effective pain management [19]. Specifically, this research evaluates the knowledge; attitudes, practices, and perceived barriers critical care nurse's face in managing pain for intubated patients in the Southern West Bank. By identifying nurse-related, physician-related, patient-related, and systemic obstacles, the study seeks to provide a foundation for targeted interventions to enhance pain management protocols and training programs. The ultimate

goal is to improve the quality of pain management, leading to better patient outcomes and satisfaction while informing health policies that address regional healthcare disparities [20].

## Methods

### Study design, sample, and setting

A descriptive cross-sectional design was used to evaluate the knowledge, attitudes, practices, and perceived barriers of critical care nurses regarding pain management for intubated patients. This design enabled data collection at a single point in time, providing a comprehensive snapshot of the targeted population's current knowledge and attitudes [15].

The study was conducted in governmental and non-governmental hospitals across the Southern West Bank, specifically in intensive care units (ICUs) and coronary care units (CCUs). These hospitals were selected due to their high patient admission rates for critically ill and intubated patients, ensuring a relevant study population. Patients in these settings typically present with complex medical conditions requiring prolonged intubation, making effective pain management a critical component of care. By selecting diverse hospital settings, the study aimed to improve the generalizability of findings to a broader range of healthcare facilities in the region [16].

### Sampling

A convenience sampling method was employed to recruit participants, targeting nurses working in the selected ICU and CCU units. The required sample size was calculated using an online sample size calculator (Raosoft) with a 95% confidence level, a 5% margin of error, and an estimated population of 300 nurses, resulting in a minimum sample of 169 participants. To account for potential non-response, the sample size was adjusted to 204 nurses, and 199 ultimately participated. Five were excluded due to being on leave or failing to return completed questionnaires [17].

#### Inclusion criteria.

- Registered nurses currently working in ICU or CCU settings

- A minimum of six months of experience in critical care

- Willingness to participate and provide informed consent

#### Exclusion criteria.

- Nurses working in non-critical care units

- Administrative nursing staff with no direct patient interaction

- Nurses on extended leave during the study period

Convenience sampling was chosen due to logistical constraints and the need for timely data collection; however, this approach may introduce selection bias. Convenience sampling, while practical, may have led to the overrepresentation of nurses who are more proactive in professional development, potentially inflating knowledge levels compared to the general ICU nurse population. Conversely, those experiencing higher workloads may have been underrepresented, skewing perceptions of time constraints and systemic barriers. Future research should consider randomized sampling techniques to improve generalizability.

### Instruments and validation

The study utilized a structured questionnaire composed of three sections to collect data:

1. **Demographic information:** Age, gender, academic qualifications, employment location, years of professional experience, and prior education related to pain management. (See S1 Questionnaire for details).

2. **Nurses' Knowledge and Attitudes Survey Regarding Pain (NKASRP):** Originally developed by Ferrell and McCaffery, this tool assesses pain management knowledge and attitudes among nurses. It was adapted to fit cultural and linguistic nuances specific to nurses in the Southern West Bank. A bilingual panel of nursing educators and pain management experts reviewed the translation for clarity and relevance [15,16]. (See S1 Questionnaire).

3. **Perceived Barriers to Pain Management Scale:** This tool, initially developed by Coker et al. and later adapted by Elcigil et al., was modified to reflect system-level challenges specific to Palestinian healthcare settings, such as limited access to specialized pain management teams and communication difficulties [16]. (See S1 Questionnaire).

To ensure validity and reliability, both instruments were pre-tested on a small sample of ICU nurses (n = 15) who were not included in the main study. Cronbach's alpha coefficients exceeded 0.70 for both instruments, confirming their internal consistency [17].

## Data collection

The researchers coordinated with hospital administrators and department heads to explain the purpose of the study and the data collection process. Nurses who met the inclusion criteria and expressed interest in participating were given detailed study information, including its objectives, procedures, potential risks, their right to withdraw, confidentiality, anonymity, and a consent form. Participants were required to sign an informed consent form before proceeding.

Data collection was conducted using a self-administered electronic questionnaire distributed between June 5, 2024, and July 15, 2024. Nurses were provided quiet rooms in their respective hospitals to complete the questionnaire, which took approximately 15 minutes. The researchers were present to address any questions and clarify survey items. Completed questionnaires were returned directly to the researchers on the same day.

## Ethical consideration

Ethical approval was secured from Palestine Ahliya University's Institutional Review Board (IRB) (Project Number: CAMS/CCNA/25/524). The study adhered to the ethical principles outlined in the Declaration of Helsinki. Participants were informed that their involvement was voluntary, and declining participation would not affect their professional standing or responsibilities.

Confidentiality and anonymity were rigorously maintained by coding the data and restricting access to the research team. Participants provided written informed consent before inclusion in the study, and all ethical and legal standards were upheld to protect participants' rights.

## Data analysis approach

The collected data were entered and analyzed using IBM SPSS Statistics version 27. Descriptive statistics summarized demographic data and questionnaire responses, including frequencies and percentages for categorical variables and means with standard deviations for continuous variables.

Inferential statistical analyses included independent t-tests and one-way ANOVA tests to examine the relationship between nurses' knowledge and attitudes with demographic

characteristics such as education level and years of experience. Pearson correlation tests explored associations between different barrier categories, including system-related, nurse-related, patient-related, and physician-related factors. Multivariate regression analysis identified predictors of knowledge levels and pain management practices.

A p-value of $< 0.05$ was considered statistically significant in all analyses. All statistical tests were two-tailed to ensure rigorous hypothesis testing and account for potential confounding variables.

## Results

### Participant demographics

Out of 204 distributed questionnaires, 199 (97.5% response rate) were completed and returned by the nurses. The demographic analysis revealed that 114 (57.3%) of participants were under 30 years old. The majority of respondents were male, comprising 110 (55.3%), and 104 (52.3%) identified as single. Most participants held a bachelor's degree in nursing (129 or 64.8%). Additionally, 147 (73.9%) had less than 5 years of experience, while 83 (41.7%) had received prior education on pain management, as shown in Table 1.

### Nurses' knowledge and attitudes regarding pain management

Analysis of the nurses' knowledge and attitudes regarding pain management indicated that only 7 (3.5%) nurses had an adequate level of knowledge, while 192 (96.5%) demonstrated inadequate knowledge, as shown in Table 2, Fig 1.

Table 1. Participant's characteristics.

| Demographic Characteristics | N | % |
| --- | --- | --- |
| Age | | |
| Less than 30 years | 114 | 57.3 |
| 30 - 39 years | 68 | 34.2 |
| 40 years or above | 17 | 8.5 |
| Gender | | |
| Male | 110 | 55.3 |
| Female | 89 | 44.7 |
| Marital Status | | |
| Single | 104 | 52.3 |
| Married | 89 | 44.7 |
| Divorced | 6 | 3.0 |
| Educational Level | | |
| Diploma | 59 | 29.6 |
| Bachelor | 129 | 64.8 |
| Master or above | 11 | 5.5 |
| Experience in ICU | | |
| Less than 5 years | 147 | 73.9 |
| 5 - 9 years | 46 | 23.1 |
| ≥ 10 years | 6 | 3.0 |
| Previous Pain Education | | |
| Yes | 83 | 41.7 |
| No | 116 | 58.3 |

**Table 2. Levels of nurses' knowledge and attitudes regarding pain.**

| Knowledge and Attitudes | N | % | M (SD) |
|---|---|---|---|
| Adequate knowledge | 7 | 3.5 | 45.7 (10.4) |
| Inadequate knowledge | 192 | 96.5 | |

*Note: N = sample size; % = percentage; cutoff point = 80%*

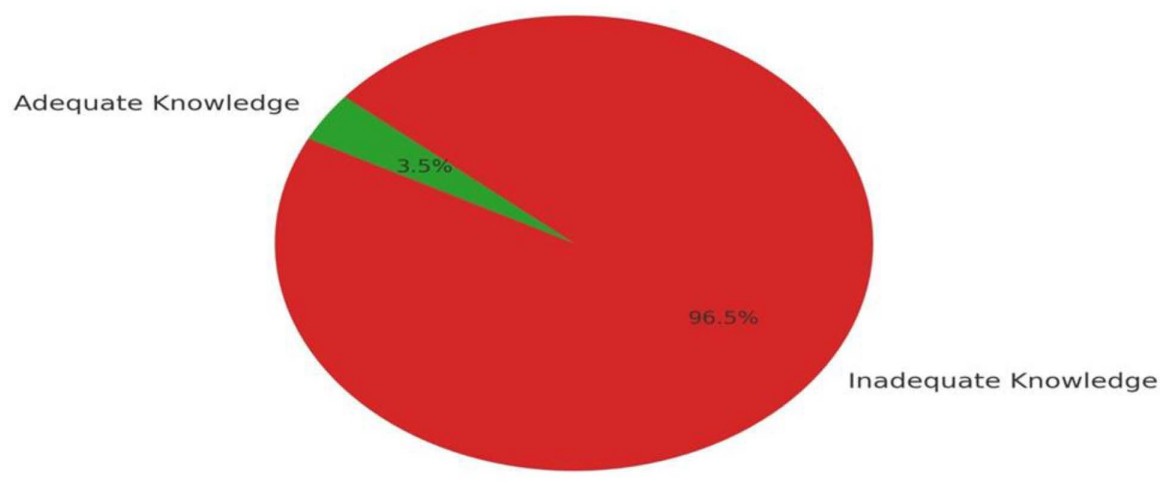

**Fig 1.** Nurses' knowledge levels on pain management.

*"I feel unsure about assessing pain in intubated patients because we were never formally trained on standardized pain scales,"* one participant noted.

*"Sometimes, even when we suspect a patient is in pain, we hesitate to administer medication due to the lack of clear protocols,"* said one ICU nurse with seven years of experience.

*"We rely on our intuition rather than standardized assessment tools, which makes pain management inconsistent,"* another nurse explained.

## Nurses' perceived barriers to pain management

The analysis revealed a mean score of 2.37 ± 0.39 for the total perceived barriers to pain management among nurses. Among the barrier types, system-related barriers were identified as the most significant obstacles to effective pain management, exhibiting the highest mean score of 2.41 ± 0.44. In contrast, physician-related barriers had the lowest mean score of 2.28 ± 0.53, as presented in Table 3. This indicates that nurses perceive systemic factors, such as a lack of standardized protocols and difficulties in communication, as more substantial challenges in managing pain effectively compared to barriers related to physician interactions.

**Table 3. Nurses' perceived barriers to management of pain.**

| Barriers | Min | Max | M | SD |
|---|---|---|---|---|
| Patients Related | 1.00 | 3.00 | 2.35 | 0.44 |
| Nurse Related | 1.00 | 3.00 | 2.40 | 0.51 |
| Physician Related | 1.00 | 3.00 | 2.28 | 0.53 |
| System Related | 1.36 | 3.00 | 2.41 | 0.44 |
| Total | 1.12 | 3.00 | 2.37 | 0.39 |

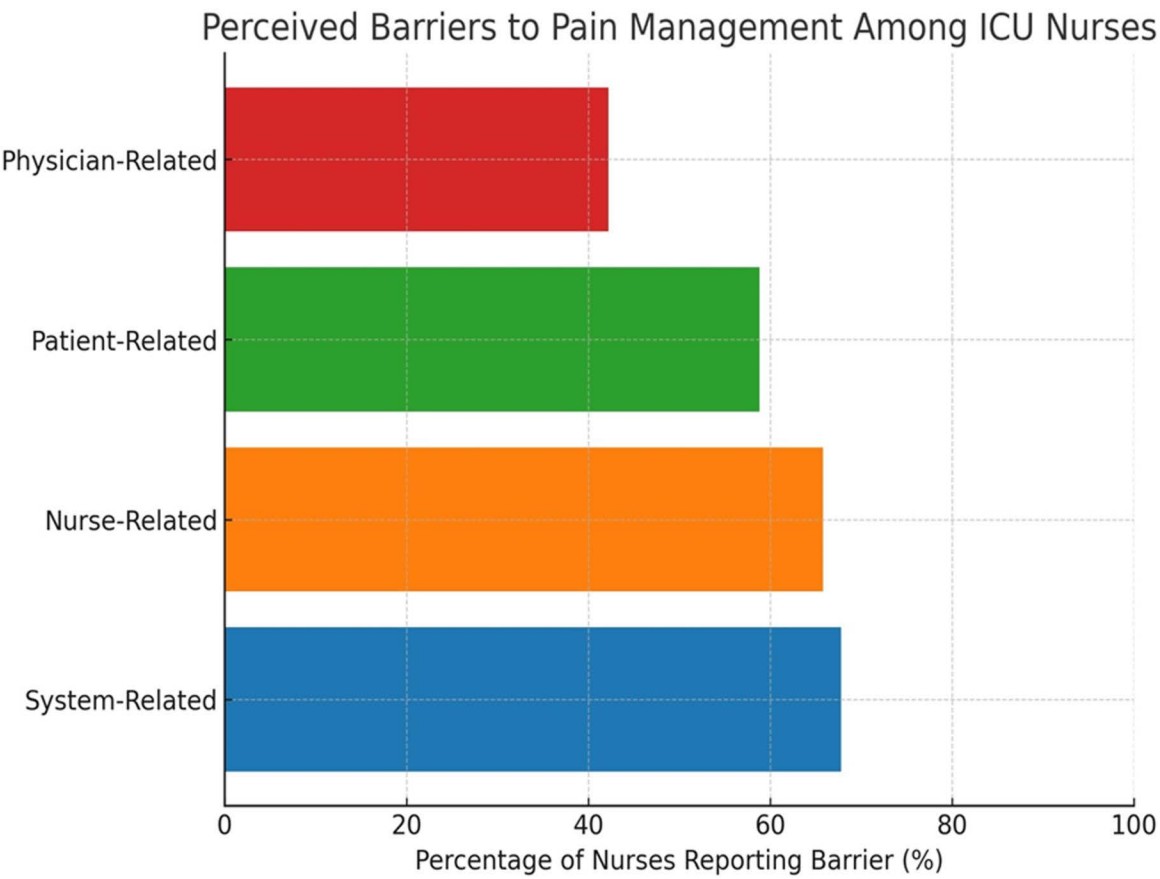

**Fig 2.** Perceived barriers to pain management among ICU nurses.

Nurses identified multiple barriers impacting effective pain management, classified into system-related, nurse-related, patient-related, and physician-related challenges. Fig 2.

One nurse explained, *"We lack clear hospital-wide guidelines, and sometimes we have to rely on our own judgment when managing pain in intubated patients."*

## Patient-related barriers

Among patients-related barriers, "Patients' difficulty with completing pain scales" (144, 72.4%) and "Patients' reluctance to report pain" (117, 58.8%) were the most commonly reported challenges. The least reported barrier was "Patients reporting their pain to the doctor, but not to the nurse" (66, 33.2%), as shown in Table 4.

## Nurse-related barriers

For nurse-related barriers, "Inadequate time for health teaching with patients" (147, 73.9%) and "Inadequate time to deliver non-pharmacologic pain relief measures" (131, 65.8%) were the most frequently reported issues. Conversely, the least reported barrier was "Inadequate staff knowledge of pain management" (87, 43.7%), as illustrated in Table 5

## Physician-related barriers

For physician-related barriers, the most frequently reported issue was "Doctor's indifference" (114, 57.3%). The least reported barriers included "Physicians' reluctance to prescribe opiates because of the side effects," "Inadequate knowledge of pain management," and "Physicians' lack of trust in the nursing assessment of pain," each reported by 84 (42.2%) nurses, as shown in Table 6.

## System-related barriers

The analysis of system-related barriers indicated that "Lack of guidelines for pain management" (135, 67.8%) and "Difficulty contacting or communicating with physicians to discuss treatment of pain in patients" (131, 65.8%) were the most commonly reported barriers. In contrast, the least reported barrier was "Inconsistent practices around giving as-needed medications for the patient" (81, 40.7%), as shown in Table 7.

## Nurses' knowledge and attitudes by demographic characteristics

To assess differences between mean scores of nurses' knowledge and attitudes regarding pain management and their demographic characteristics, independent t-tests and ANOVA tests

**Table 4. Patients-related barriers.**

| Patients-related Barriers | Agree | Disagree | Do Not Know |
|---|---|---|---|
| Patients' difficulty with completing pain scales (e.g., 0-10) | 144 (72.4) | 40 (20.1) | 15 (7.5) |
| Consumers not demanding results | 111 (55.8) | 66 (33.2) | 22 (11.1) |
| Patients' reluctance to take pain medication for fear of addiction | 108 (54.3) | 55 (27.6) | 36 (18.1) |
| Caregiver's indifference | 80 (40.2) | 79 (39.7) | 40 (20.1) |
| Patients' reluctance to take pain medications because of side effects | 99 (49.7) | 53 (26.6) | 47 (23.6) |
| Patients reporting their pain to the doctor, but not to the nurse | 66 (33.2) | 100 (50.3) | 33 (16.6) |
| Patients' reluctance to take opioids | 88 (44.2) | 69 (34.7) | 42 (21.1) |
| Patients' reluctance to report pain | 117 (58.8) | 56 (28.1) | 26 (13.1) |
| Patients not wanting to bother the nurses | 105 (52.8) | 72 (36.2) | 22 (11.1) |

**Table 5. Nurses-related barriers.**

| Nurses-related Barriers | Agree | Disagree | Do Not Know |
|---|---|---|---|
| Inadequate time for health teaching with patients | 147 (73.9) | 25 (12.6) | 27 (13.6) |
| Inadequate time to deliver non-pharmacologic pain relief measures | 131 (65.8) | 19 (9.5) | 49 (24.6) |
| Inadequate staff knowledge of pain management | 87 (43.7) | 92 (46.2) | 20 (10.1) |
| Nursing staff reluctance to administer opiates | 90 (45.2) | 79 (39.7) | 30 (15.1) |
| Fear of pain medications because of side effects | 124 (62.3) | 35 (17.6) | 40 (20.1) |
| Inadequate assessment of pain | 96 (48.2) | 74 (37.2) | 29 (14.6) |
| Nurses' indifference | 105 (52.8) | 62 (31.2) | 32 (16.1) |

Table 6. Physician-related barriers.

| Physicians-related Barriers | Agree | Disagree | Do Not Know |
|---|---|---|---|
| Inadequate assessment of pain and pain relief | 104 (52.3) | 66 (33.2) | 29 (14.6) |
| Doctor's indifference | 114 (57.3) | 55 (27.6) | 30 (15.1) |
| Physicians' reluctance to prescribe opiates because of the side effects | 84 (42.2) | 51 (25.6) | 64 (32.2) |
| Inadequate knowledge of pain management | 84 (42.2) | 92 (46.2) | 23 (11.6) |
| Physicians' fear of addiction of medicine | 98 (49.2) | 54 (27.1) | 47 (23.6) |
| Physicians' reluctance to prescribe adequate pain relief for fear of overmedicating | 104 (52.3) | 39 (19.6) | 56 (28.1) |
| Physicians' lack of trust in the nursing assessment of pain | 84 (42.2) | 81 (40.7) | 34 (17.1) |

Table 7. System-related barriers.

| System-related Barriers | Agree | Disagree | Do Not Know |
|---|---|---|---|
| Lack of psychosocial support services | 117 (58.8) | 41 (20.6) | 41 (20.6) |
| Patient-to-nurse ratio | 115 (57.8) | 34 (17.1) | 50 (25.1) |
| Lack of social workers experienced in hospital settings | 99 (49.7) | 63 (31.7) | 37 (18.6) |
| Lack of guidelines for pain management | 135 (67.8) | 38 (19.1) | 26 (13.1) |
| Lack of access to professionals who practice specialized pain treatment methods | 118 (59.3) | 54 (27.1) | 27 (13.6) |
| Difficulty contacting or communicating with physicians to discuss treatment of pain in patients | 131 (65.8) | 38 (19.1) | 30 (15.1) |
| Not having a documented pain treatment plan for each patient | 122 (61.3) | 38 (19.1) | 39 (19.6) |
| Lack of alternative non-pharmacologic therapy for pain management | 122 (61.3) | 52 (26.1) | 25 (12.6) |
| Inconsistent practices around giving as-needed medications for patients | 81 (40.7) | 51 (25.6) | 67 (33.7) |
| Lack of medicine in markets | 116 (58.3) | 70 (35.2) | 13 (6.5) |
| Lack of equipment or skill in using equipment | 115 (57.8) | 70 (35.2) | 14 (7.0) |

were conducted. The analysis revealed significant differences according to educational level ($P < 0.05$) and previous pain education ($P < 0.05$). Nurses who had received previous pain education had a higher mean score (M = 48.1 ± 13.5) compared to those who had not (M = 44.0 ± 7.0), as summarized in Table 8.

## Discussion

This study examined the knowledge, attitudes, practices, and perceived barriers of critical care nurses regarding pain management for intubated patients in Southern West Bank hospitals. By utilizing the Nurses' Knowledge and Attitudes Survey Regarding Pain, valuable insights were gained. Our findings reveal that a vast majority of nurses (96.5%) demonstrated inadequate knowledge of pain management, a result that is consistent with previous regional studies [19]. Similar research conducted in neighboring countries, including Jordan and Lebanon, has reported comparable challenges related to insufficient training and the lack of standardized protocols [20].

The analysis also highlighted several barriers impacting effective pain management. System-related barriers, such as the absence of clear, standardized guidelines and difficulties in nurse–physician communication, were identified as major challenges. These findings echo earlier reports from the region [19]. Additionally, nurse-related and patient-related barriers were observed, further complicating the delivery of optimal pain management practices [19,20].

The results reveal a substantial gap in knowledge among critical care nurses, with a notable percentage exhibiting insufficient understanding of pain management. This gap is influenced by multiple factors, including limited opportunities for professional development

**Table 8. Differences in knowledge and attitudes by demographics.**

| Variable | M (SD) | Statistical Test | p-value |
|---|---|---|---|
| Gender | | | |
| Male | 46.1 (9.5) | t = 0.548 | .584 |
| Female | 45.3 (11.4) | | |
| Educational Level | | F = 3.703 | .026* |
| Diploma | 8.2 (1.1) | | |
| Bachelor | 10.9 (1.0) | | |
| Master or above | 11.4 (0.4) | | |
| Experience | | F = 2.614 | .076 |
| Less than 5 years | 46.7 (11.2) | | |
| 5 - 9 years | 42.9 (7.3) | | |
| ≥ 10 years | 42.8 (5.5) | | |
| Age | | F = 2.813 | .062 |
| Less than 30 years | 47.2 (11.7) | | |
| 30 - 39 years | 43.8 (8.6) | | |
| 40 years or above | 43.5 (4.7) | | |
| Previous Pain Education | | t = 2.50 | 0.014* |
| Yes | 48.1 (13.5) | | |
| No | 44.0 (7.0) | | |

*Significant at the 0.05 level.

and the inconsistent incorporation of pain management education within nursing curricula [17,18,19]. Similar trends have been reported in global studies, where critical care nurses in low-resource settings demonstrate significant deficiencies in pain management knowledge due to inadequate formal training and limited access to continuing education opportunities [1,6,20].

Attitudinal barriers were identified as significant factors impacting practices. The study indicated that nurses often underestimate or dismiss patients' pain reports due to misconceptions about pain expression, leading to underassessment and inadequate pain relief. This finding echoes earlier research highlighting negative attitudes toward self-reports as a barrier to effective pain control [20]. A recent global review found that cultural perceptions of pain, combined with high workloads and stress, contribute to reluctance in administering adequate analgesia, even when clinical evidence supports its necessity [2]. Such attitudes may also stem from workplace stress and decision-making challenges nurses face in high-pressure environments, as highlighted in recent studies [17]. Addressing these attitudinal challenges is critical for improving the quality of pain management and enhancing patient outcomes [21].

Analysis of perceived barriers revealed significant systemic issues, including the absence of standardized protocols and communication difficulties, as the primary challenges to effective pain management. These systemic obstacles reflect findings from the literature, emphasizing the necessity for organizational support and the establishment of clear pain management protocols [5,7]. The impact of systemic barriers is further reinforced by recent evidence highlighting that ICUs without well-defined pain management policies often exhibit lower adherence to evidence-based practices and increased variability in patient outcomes [4].

Patient-related challenges, such as difficulties in using pain scales and hesitance to report pain, were also noted. The results indicate that nurses are aware of these issues, highlighting the need for alternative pain assessment tools that cater to non-verbal patients [22]. Emerging

research suggests that implementing artificial intelligence (AI)-based pain assessment tools in ICUs could improve the accuracy of pain detection in intubated patients, particularly when traditional assessment methods are ineffective [6].

Additionally, nurse-related barriers, including insufficient time for education and the delivery of non-pharmacologic measures, reflect the heavy workload encountered by nurses in critical care environments. These challenges are compounded by stress and burnout, which have been found to influence resilience and decision-making under challenging conditions [23]. This finding is consistent with the literature that emphasizes the influence of time constraints and psychological stress on effective pain management [24]. A 2023 study reported that integrating digital training modules into nursing shifts can mitigate these barriers by providing on-demand education without disrupting patient care responsibilities [25].

The findings concerning physician-related barriers, such as hesitancy to prescribe opioids and insufficient knowledge about pain management, are consistent with literature that points to similar challenges faced by healthcare providers globally [26,27]. Physicians' reluctance to prescribe adequate analgesia, often due to concerns about opioid dependency and regulatory restrictions, remains a major issue, as documented in international pain management guidelines [28]. Enhancing communication and education across disciplines regarding pain management practices is vital for overcoming these barriers.

The findings of this study not only highlight the barriers faced in the Southern West Bank but also have implications for global healthcare. Similar challenges are observed in other middle- and low-income countries, where resource limitations, staffing shortages, and inconsistent policies affect pain management quality. Studies from sub-Saharan Africa and Southeast Asia have reported nearly identical barriers, emphasizing the global need for structured pain management protocols and interdisciplinary collaboration [29].

## Strengths and limitations

The strengths of the study include a high response rate (97.5%), which enhances the reliability of the findings, and the use of validated instruments for measuring knowledge and attitudes. Its focus on pain management for intubated patients addresses an important aspect of nursing practice.

However, the study's cross-sectional design limits the ability to establish causal relationships, and the findings may not be generalizable to other contexts. Additionally, the reliance on self-reported data may introduce bias, affecting the accuracy of the results. Future research should consider longitudinal studies to assess the long-term effects of pain management interventions and training programs.

## Recommendations

To address these barriers, several innovative and region-specific recommendations can be considered. Hospitals should adopt standardized pain management protocols to ensure uniform, evidence-based guidelines that reduce discrepancies in practice and promote adherence to best practices. Targeted continuing education should be integrated into nurses' professional development, incorporating mandatory training programs, digital learning modules, and simulation-based education to reinforce practical skills without disrupting workflow. Enhancing interdisciplinary collaboration through regular workshops between physicians, nurses, and pharmacists can improve communication and minimize physician-related barriers to pain treatment. Telehealth for pain management training can provide nurses with virtual education programs and online certification courses, overcoming geographic and scheduling constraints. The implementation of AI-powered pain assessment

tools in ICUs can enhance the accuracy of pain detection for non-verbal patients, particularly those under mechanical ventilation. Additionally, mobile-based pain assessment applications can assist nurses with real-time, evidence-based bedside pain evaluation, supporting more efficient clinical decision-making. Institutional policy reform is essential, with government and hospital administrators integrating mandatory pain management education into nursing curricula and ongoing professional development programs to ensure sustainable improvements in pain management practices. By implementing these recommendations, hospitals and policymakers can significantly enhance pain management practices, improve patient outcomes, and create a more supportive work environment for healthcare providers. In conclusion, this study underscores the urgent need for comprehensive reforms in pain management within critical care settings. The consistency of our findings with those of other regional studies [19,20] further highlights the importance of adopting these targeted strategies to achieve meaningful improvements in patient care.

### Directions for future research

Future research should focus on developing and evaluating evidence-based educational interventions to improve nurses' knowledge and attitudes toward pain management, particularly in resource-limited settings. Studies exploring the effectiveness of AI-driven pain assessment tools for non-verbal ICU patients could provide valuable insights into technology-assisted pain management. Additionally, longitudinal studies assessing the impact of standardized pain management protocols on patient outcomes and nursing practices would help establish best practices tailored to critical care settings. Further research is also needed to investigate the role of interdisciplinary collaboration in overcoming physician-related barriers and improving pain management strategies. Qualitative studies exploring nurses' experiences, emotional burdens, and ethical dilemmas related to pain management could offer deeper insights into attitudinal barriers and potential interventions. Finally, policy-oriented research examining the integration of pain management training into nursing curricula and continuing education programs would help guide institutional and governmental reforms, ensuring a sustained improvement in pain management practices.

### Conclusion

This study identified significant gaps in critical care nurses' knowledge, attitudes, practices, and perceived barriers to effective pain management for intubated patients in Southern West Bank hospitals. Alarmingly, 96.5% of nurses exhibited inadequate knowledge, highlighting the urgent need for enhanced education. Systemic and attitudinal barriers, particularly the absence of standardized pain management protocols and negative perceptions toward pain treatment, must be addressed through targeted interventions.

To improve patient care, healthcare institutions should implement standardized pain management guidelines, enhance interdisciplinary communication, and integrate ongoing pain education into nursing curricula. Policymakers must also invest in accessible professional development opportunities, such as telehealth-based training and AI-driven assessment tools. By taking these steps, critical care environments can improve pain management outcomes, ultimately benefiting patient recovery and well-being.

### Supporting information

**S1 Questionnaire. Study questionnaire.**
(DOCX)

## Acknowledgements

The authors would like to express their thanks to the nurses who participated in the study

## Author contributions

**Conceptualization:** Ibrahim Salim, Ahmad Ayed, Ibrahim Aqtam.

**Data curation:** Ibrahim Salim, Ahmad Ayed.

**Formal analysis:** Moath Abu Ejheisheh, Ibrahim Aqtam.

**Investigation:** Ibrahim Salim.

**Methodology:** Ibrahim Salim, Moath Abu Ejheisheh, Ibrahim Aqtam.

**Project administration:** Ibrahim Salim.

**Supervision:** Ahmad Batran.

**Writing – original draft:** Ibrahim Salim, Ahmad Ayed.

**Writing – review & editing:** Moath Abu Ejheisheh, Ibrahim Aqtam, Ahmad Batran.

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
