## [Decision Letter · Decision Letter 0]

11 Feb 2025

PONE-D-25-01292Knowledge, Attitudes, Practice and Perceived Barriers of Critical Care Nurses Regarding Pain Management for Intubated Patients in Southern West Bank Hospitals.PLOS ONE

Dear Dr. aqtam, Thank you for submitting your manuscript to PLOS ONE. After careful consideration, we feel that it has merit but does not fully meet PLOS ONE’s publication criteria as it currently stands. Therefore, we invite you to submit a revised version of the manuscript that addresses the points raised during the review process.

We look forward to receiving your revised manuscript.

Kind regards,

Othman A. Alfuqaha, Ph.D.

Academic Editor

PLOS ONE

Journal Requirements:

2.  Peer review at PLOS ONE is not double-blinded (https://journals.plos.org/plosone/s/editorial-and-peer-review-process). For this reason, authors should include in the revised manuscript all the information removed for blind review.

3. Please include captions for your Supporting Information files at the end of your manuscript, and update any in-text citations to match accordingly. Please see our Supporting Information guidelines for more information: http://journals.plos.org/plosone/s/supporting-information .

**Additional Editor Comments:**

The manuscript has significant merit but requires clarification and methodological refinements before acceptance. Please follow my comments as well as reviewer comments.

1. Title Revision: Please, refine the title to be more concise and engaging while maintaining clarity.

2. Discussion Enhancement: Please compare your findings more explicitly with international pain management guidelines and discuss the broader implications.

3. Methodological Justification. Please more detsailas are needed regarding your methods. Be aware to answer 5 W questions in your method section.

4. Address the selection bias concern, either by acknowledging its limitations or justifying why the sample is still representative.

5. The response rate calculation should be corrected based on the total number of critical care nurses in participating facilities.

6. A clear explanation of subgroup analyses should be added to the Methods section.

7. Generalizability Consideration: Discuss the limitations introduced by selection bias and suggest how future studies could enhance external validity.

Reviewers' comments:

Reviewer's Responses to Questions

**Comments to the Author**

1. Is the manuscript technically sound, and do the data support the conclusions?

Reviewer #1: Yes

Reviewer #2: No

2. Has the statistical analysis been performed appropriately and rigorously?

Reviewer #1: Yes

Reviewer #2: Yes

3. Have the authors made all data underlying the findings in their manuscript fully available?

Reviewer #1: Yes

Reviewer #2: Yes

4. Is the manuscript presented in an intelligible fashion and written in standard English?

Reviewer #1: Yes

Reviewer #2: Yes

5. Review Comments to the Author

Reviewer #1: it was good articles need some minor changes as it was attached , Clarity in Title: The title is too descriptive; it needs to be shortened to make it more readable the findings of the study were set against the backdrop of global trends in pain management in intubated patients to establish a broader importance of the study. Discuss deeper the comparison of the identified systemic and attitudinal barriers in relation to international standards or guidelines on pain management.

Reviewer #2: Pain management is a critical aspect of healthcare, and research aimed at achieving high-quality pain management is equally essential. The authors investigated nurses' knowledge, attitudes, practices, and perceived barriers regarding pain management. These factors constitute vital information for delivering high-quality pain management. However, there are several methodological concerns that warrant consideration.

Major Comment

This study employed convenience sampling, and it can be inferred that the questionnaires were predominantly distributed to nurses with a particular interest in pain management. Consequently, this sampling method may have introduced selection bias, potentially affecting the generalizability of the findings.

Given that the target population consists of critical care nurses, the denominator for calculating the response rate should be the total number of critical care nurses in the participating facilities rather than the number of distributed questionnaires.

Minor comments

All subgroup analyses performed should be explicitly described in the Methods section, including the rationale for the analyses and the specific statistical methods employed.

6. PLOS authors have the option to publish the peer review history of their article (what does this mean? ). If published, this will include your full peer review and any attached files.

**Do you want your identity to be public for this peer review?** For information about this choice, including consent withdrawal, please see our Privacy Policy .

Reviewer #1: No

Reviewer #2: **Yes: ** Takeshi Unoki, PhD

---

## [Author Response · Author response to Decision Letter 1]

11 Feb 2025

Dear Editor and Reviewers,

We sincerely appreciate the time and effort you have taken to review our manuscript. We provide a point-by-point response to each comment. We have revised the manuscript accordingly and highlighted changes in the revised document (uploaded as "Revised Manuscript with Track Changes").

---

## [Decision Letter · Decision Letter 1]

17 Feb 2025

PONE-D-25-01292R1Barriers and Practices in Pain Management for Intubated Patients: A Study of Critical Care Nurses in Southern West Bank HospitalsPLOS ONE

Dear Dr. aqtam, Thank you for submitting your manuscript to PLOS ONE. After careful consideration, we feel that it has merit but does not fully meet PLOS ONE’s publication criteria as it currently stands. Therefore, we invite you to submit a revised version of the manuscript that addresses the points raised during the review process.

We look forward to receiving your revised manuscript.

Kind regards,

Othman A. Alfuqaha, Ph.D.

Academic Editor

PLOS ONE

Journal Requirements:

Additional Editor Comments:

Thank you for your revised manuscript to PLoS ONE. Before proceeding with the acceptance of your manuscript, we kindly request that you carefully address the reviewers’ comments and provide a point-by-point response to their feedback. Ensuring that all concerns are adequately addressed will strengthen your work and enhance its contribution to the field.

Reviewers' comments:

Reviewer's Responses to Questions

**Comments to the Author**

1. If the authors have adequately addressed your comments raised in a previous round of review and you feel that this manuscript is now acceptable for publication, you may indicate that here to bypass the “Comments to the Author” section, enter your conflict of interest statement in the “Confidential to Editor” section, and submit your "Accept" recommendation.

Reviewer #1: All comments have been addressed

Reviewer #2: (No Response)

2. Is the manuscript technically sound, and do the data support the conclusions?

Reviewer #1: Yes

Reviewer #2: Partly

3. Has the statistical analysis been performed appropriately and rigorously?

Reviewer #1: Yes

Reviewer #2: Yes

4. Have the authors made all data underlying the findings in their manuscript fully available?

Reviewer #1: Yes

Reviewer #2: Yes

5. Is the manuscript presented in an intelligible fashion and written in standard English?

Reviewer #1: Yes

Reviewer #2: Yes

6. Review Comments to the Author

Reviewer #1: The study provides valuable insights into pain management practices for intubated patients in critical care settings. To enhance clarity, the introduction could better highlight the study's gaps in existing literature. Additionally, the methodology section would benefit from further explanation of the sample's inclusion/exclusion criteria and the data analysis approach. The results section is well-organized, but incorporating more visual aids and direct quotes from participants would strengthen it. In the discussion, a comparison with other regional studies and more detailed practical recommendations for overcoming identified barriers would be beneficial. Finally, including ethical considerations and directions for future research would enhance the manuscript’s overall impact.

Reviewer #2: While some aspects have been revised, the discussion of response rates and the impact of selection bias remains unclear.

The study appears to capture data primarily from critical care nurses with a specific interest in pain management. This raises concerns about whether your findings truly represent the broader population of hospital critical care nurses. The potential self-selection bias suggests that your sample may not adequately reflect the perspectives, practices, and attitudes of the general critical care nursing population.

The authors should clearly state:

1.the total number of critical care nurses in the population

2.the number of surveys distributed

3.the number of surveys returned

Furthermore, they should thoroughly examine how selection bias may have influenced their findings. The potential impact of selection bias needs to be adequately addressed in the discussion.

7. PLOS authors have the option to publish the peer review history of their article (what does this mean? ). If published, this will include your full peer review and any attached files.

**Do you want your identity to be public for this peer review?** For information about this choice, including consent withdrawal, please see our Privacy Policy .

Reviewer #1: No

Reviewer #2: **Yes: ** Takeshi Unoki, PhD

---

## [Author Response · Author response to Decision Letter 2]

17 Feb 2025

Please find our point‐by‐point responses: We trust that these revisions and clarifications address the reviewers’ and editor’s concerns. We appreciate the opportunity to improve our manuscript and look forward to your further feedback.

---

## [Editor Report · Decision Letter 2]

19 Feb 2025

Barriers and Practices in Pain Management for Intubated Patients: A Study of Critical Care Nurses in Southern West Bank Hospitals

PONE-D-25-01292R2

Dear Dr. Ibrahim,

We’re pleased to inform you that your manuscript has been judged scientifically suitable for publication and will be formally accepted for publication once it meets all outstanding technical requirements.

Kind regards,

Othman A. Alfuqaha, Ph.D.

Academic Editor

PLOS ONE

Additional Editor Comments (optional):

Dear authors, Im satisfied with the revision process, and hence I recommend approving this paper for publication. Thanks for your valuable work and best of luck.
---

## [Editor Report · Acceptance letter]

PONE-D-25-01292R2

PLOS ONE

Dear Dr. aqtam,

I'm pleased to inform you that your manuscript has been deemed suitable for publication in PLOS ONE. Congratulations! Your manuscript is now being handed over to our production team.

Kind regards,

on behalf of

Dr. Othman A. Alfuqaha

Academic Editor

PLOS ONE